# Ejaculations and Benign Prostatic Hyperplasia: An Impossible Compromise? A Comprehensive Review

**DOI:** 10.3390/jcm10245788

**Published:** 2021-12-10

**Authors:** Nicolas Couteau, Igor Duquesne, Panthier Frédéric, Nicolas Thiounn, Marc-Olivier Timsit, Arnaud Mejean, Ugo Pinar, François Audenet

**Affiliations:** 1Service d’Urologie, Hôpital Européen Georges Pompidou, Université de Paris, 75015 Paris, France; nicolas.couteau@hotmail.com (N.C.); nicolas.thiounn@aphp.fr (N.T.); marc-olivier.timsit@aphp.fr (M.-O.T.); arnaud.mejean@aphp.fr (A.M.); francois.audenet@aphp.fr (F.A.); 2Service d’Urologie, Hôpital Cochin, Université de Paris, 75014 Paris, France; igor.duquesne@aphp.fr; 3Service d’Urologie, Hôpital La Pitié-Salpétriere, Sorbonne Université, 75013 Paris, France; ugo.pinar@aphp.fr

**Keywords:** benign prostatic hyperplasia, ejaculation, endoscopic enucleation, anatomy, ejaculation disorders

## Abstract

Background: Benign prostatic hyperplasia (BPH) is commonly responsible for lower urinary tract symptoms (LUTS) in men aged 50 or over. Sexual dysfunctions, such as ejaculatory disorders (EjD), go along with LUTS but are frequently overlooked in the initial evaluation. This review aimed to detail BPH-related EjD, as well as their modifications by medical, surgical, and interventional treatments. Methods: We conducted a narrative review looking for publications between 1990 and 2020, regarding physiopathology, epidemiology, evaluation, and therapeutic management (medical, surgical, and interventional) of BPH-related EjD. Results: Sixty-five articles were included in our final analysis. Forty-six percent of men presenting with LUTS reported EjD. If the prevalence increases with age and LUTS severity, the functional impairment is not correlated with age. Several self-questionnaires evaluated the sexual function, but only four approaches are specific to EjD. Medical therapies were exposed to anejaculation, rather than retrograde ejaculation (RE) (4–30% (alpha-blockers), 4–18% (5-alpha-reductase inhibitors)). Regarding surgical therapies, trans-urethral resection of the prostate (TURP) and incision of the prostate (TUIP) are associated with 50–70% and 21–35% of RE. The RE rate is important after open simple prostatectomy but can be reduced with robotic approaches and urethral sparing techniques (19%). Anatomic endoscopic enucleation of the prostate (AEEP) with or without a laser source is associated with an 11–36% RE rate, according to supramontanal preservation. Recent surgical techniques (Rezum©, Aquablation©, or Urolift©) were developed to preserve antegrade ejaculation with promising short-term results. Regardless of the surgical approach, anatomic studies suggest that the preservation of peri-montanal tissue (7.5 mm laterally; 10 mm proximally) is primordial to avoid post-operative RE. Finally, prostate artery embolization (PAE) limits the RE rate but exposes it to a 12 months 10% re-intervention rate. Conclusion: EjD concerns almost half of the patients presenting BPH-related LUTS. Initial evaluation of EjD impairment is primordial before medical or surgical therapy. Peri-montanal tissue preservation represents a key point for antegrade ejaculation preservation, regardless of the surgical option.

## 1. Introduction

The prostatic gland plays a central role in andrology. It is involved both in fertility and in sexuality with a major role in ejaculation and possibly in orgasm. This could explain the association between the andrological symptoms and prostatic disorders. Benign prostatic hyperplasia (BPH) is a common urologic condition responsible for lower urinary tract symptoms (LUTS) in nearly 80% of men aged 50 or over [1]. While LUTS are central in the clinical evaluation of symptomatic BPH, related sexual disorders are often overlooked despite a close demonstrated association [2,3]. Within these sexual modifications, ejaculatory disorders linked to BPH are rarely evaluated specifically. However, the pathophysiology of BPH-related dysejaculation is not clearly elucidated and consequently the impact of medical and surgical BPH treatments [4,5]. Symptoms related to an ejaculatory disorder are quite diverse as patients may describe pain or discomfort at the time of ejaculation, quantity or quality disorder, premature or delayed ejaculation, or retrograde ejaculation or anejaculation. Several subjective questionnaires have tried to assess these symptoms. The aim of this narrative review is to specify BPH-related ejaculatory dysfunction (EjD), as well as their modifications by medical, surgical, and interventional treatments.

## 2. Materials and Methods

### 2.1. Search Strategy

We conducted a comprehensive review of the sexual dysfunction in men with benign prostatic hyperplasia (BPH) with a specific focus on ejaculatory disorder. PubMed Medline, Cochrane, and Scopus databases were used in September 2020. The search strategy was unrestricted and used explored MeSH (medical subject heading) terms such as: “Prostatic Hyperplasia”, “lower urinary tract symptoms”, “sexual dysfunction”, and “ejaculatory dysfunction”.

### 2.2. Study Eligibility

We included articles published between January 1990 and December 2020. The literature search was limited to English-language and French-language articles. Non-human studies, clinical case reports, other-languages articles, editorials, letters, congress communications, and articles dealing with different subjects were excluded.

### 2.3. Inclusion Criteria

A review protocol guided by the Preferred Reporting Items for Systematic Reviews and Meta-Analyses (PRISMA) checklist was established following inclusion criteria for article selection: publications had to be written in English or French language, including prospective and retrospective trials providing data on physiopathology, epidemiology, evaluation and therapeutic management of ejaculatory dysfunction related to BPH. Abstracts were read by authors (FP, ID, UP, NC). Article quality was assessed on clinical interest and on the quality of the described cohorts.

Regarding the impact of surgical treatments, we considered all the existing techniques (TURP, endoscopic or open enucleation of the prostate) without excluding new techniques currently under evaluation. We also included trials assessing non-surgical invasive treatment such as prostatic arteries embolization.

A list of 65 relevant articles was selected and retrieved for further qualitative analysis.

### 2.4. Data Management

From the selected articles, the number of patients, the study design (prospective or retrospective), and the methodology to assess postoperative outcomes were extracted and analyzed. Finally, the postoperative outcomes were assessed. The quality and the heterogeneity of the included studies were evaluated based on the demographic characteristics of the population (number of patients and inclusion criteria), the study design (prospective or retrospective), and the methodology used to assess postoperative outcomes (functional, ejaculatory status).

## 3. Results

We identified 1002 citations of which 251 required full-text review after title and abstract screening, and 65 studies met inclusion criteria for inclusion in this review.

### 3.1. Physiopathology of Ejaculatory Disorders in BPH

Ejaculation is a complex phenomenon with two phases: emission and expulsion. In the emission phase, there are contractions of the deferential ampullae, the seminal vesicles, and the prostate. The bladder neck and the striated urethral sphincter close. This phase is mediated by sympathetic innervation (T10-L2). The expulsion phase includes the opening of the striated urethral sphincter associated with the persistence of prostatic contractions and the initiation of spasmodic contractions of the perineal muscles to expel sperm through the urethral meatus. This phase is mediated by sacral somatic innervation (S2–S4) [6]. The pathophysiological mechanisms of ejaculatory disorders seem related to those of erectile dysfunction and LUTS [2,5]. In the literature, four main mechanisms are mentioned: alteration of the NO-GMPc signaling pathway, hyperactivation of the RhoA-ROCK signaling channel, hyperactivation of the autonomic nervous system, and pelvic vascular arteriosclerosis.

#### 3.1.1. Alteration of the NO-GMPc Signaling Pathway

Tissues of the lower urinary tract (bladder, prostate, urethra) have been shown to contain calcium-dependent NO synthase (NOS) [2,7]. Their stimulation causes a release of nitric oxide (NO) that stimulates the cGMP, thus inducing muscle and glandular relaxation. In cases of a nerve or endothelial structures’ alteration (with impaired NO production), LUTS and symptoms of sexual dysfunction have been reported. Furthermore, an NO-GMPc track dysfunction with disturbances in the relaxation/contraction of smooth muscle cells could play a major role in the occurrence of retrograde ejaculation (bladder neck) or anejaculation (prostate).

#### 3.1.2. Hyperactivation of the RhoA-ROCK Signaling Channel

ROCK is a serine-threonine kinase that is involved in regulating the shape and movement of cells by acting on the cytoskeleton [5,8]. This pathway includes effects on smooth muscle cells of the lower urinary tract (bladder, prostate). Its hyperactivation in animal models (rat) was associated with a higher rate of development of BPH, overactive bladder and high blood pressure, and glaucoma. 

#### 3.1.3. Hyperactivation of the Autonomic Nervous System

The autonomic nervous system balances sympathetic and parasympathetic signals, regulating the emission phase in particular [8]. Therefore, any disruption to this balance should have an impact on ejaculatory function.

#### 3.1.4. Pelvic Vascular Arteriosclerosis

Finally, arteriosclerosis of the bladder, prostate, and cavernous arteries represents a significant component in the pathophysiology of ejaculatory disorders, through the other pathways [5].

### 3.2. Assessment Methods for Ejaculatory Disorders in BPH

Male sexual disorders are commonly assessed by subjective self-reports in clinical practice [9]. Several scores evaluating the sexual or erectile function are available, but only a few are dedicated to the ejaculatory status. As part of them, the Male Sexual Health Questionnaire (MSHQ), the Danish Prostatic Symptoms Score (DAN-PSS), the International Continence Society Sex (ICS-Sex), and the Brief Male Sexual Function Inventory (BMFSI) specifically evaluate ejaculatory function.

#### 3.2.1. Male Sexual Health Questionnaire (MSHQ)

This questionnaire is composed of 25 questions treating erection, ejaculation, orgasm, desire, and satisfaction of the man’s sexuality. It is the most relevant questionnaire for the evaluation of ejaculation. A sub-part of this questionnaire is dedicated to the evaluation of ejaculation (MSHQ-EjD, four items) [10].

#### 3.2.2. Danish Prostatic Symptoms Score (DAN-PSS)

To the original 12-question questionnaire were added questions about male sexual function: erection, ejaculation, ejaculatory pain, or discomfort. This score, used in particular for the MSAM-7 study, specifically covers ejaculatory function with the addition of a severity indicator [3].

#### 3.2.3. International Continence Society Sex (ICS-Sex)

This questionnaire is a sub-part of the ICS assessing BPH (ICS-PHB) [11]. Four questions of this questionnaire evaluate the impact of BPH on sexual function. They assess erections, ejaculations, ejaculatory discomfort, and their functional disorder.

#### 3.2.4. Brief Male Sexual Function Inventory (BMFSI)

The BMFSI is composed of 11 questions. Overall, 25% of the questions are about ejaculatory dysfunction [12,13].

### 3.3. Epidemiology of Ejaculatory Disorders (EjD) in BPH

Several epidemiological studies have specifically evaluated sexual or ejaculatory function in relation to BPH-related LUTS [14]. The MSAM-7 study was conducted among 34,800 men aged between 50 and 80 years old, of whom 14,254 replies were received (12,815 usable responses) [3]. Overall, 90% of the respondents reported having LUTS and 83% had sexual activity, of which 71% reported at least one sexual activity in the last four weeks. Reported sexual dysfunction and discomfort were highly correlated with age and LUTS severity, but this correlation was independent of cardiovascular comorbidities and diabetes. This study concluded that sexual dysfunction and LUTS were associated. Moreover, 46.2% of the men had reduced or absent ejaculate volume (5%) despite the fact that 86.7% and 81.3% had erections and ejaculations respectively. The prevalence of ejaculatory dysfunction increased significantly with patient age (30.1%, 54.9%, 74.4% of men aged 50–59, 60–69, and 70–80 years, respectively) and with the severity of LUTS (41.8%, 61.4%, 76% of men with mild, moderate, and severe LUTS). Functional discomfort related to ejaculatory dysfunction was not correlated with age but was significantly related to the LUTS severity. Finally, 7.2% of respondents reported ejaculatory pain or ejaculatory inconvenience but functional discomfort was substantial (88.3%). Ejaculatory pain or discomfort prevalence was significantly correlated with age and LUTS severity.

The EpiLUTS study included 11,834 men with an average age of 56 years in the statistical analysis [1]. Ejaculatory function was assessed by the MSHQ-EjD. About 71% of patients described ejaculations at every orgasm, 18% most of the time, only 6.7% described ejaculatory dysfunction (decreased volume or no ejaculation). For 47% of patients with multiple LUTS, ejaculatory dysfunction occurred at least once within the last 4 weeks. Anejaculation increased with patient age (40–45 years: 1.2%, 46–50 years: 1.4%, 51–55 years: 2%, 56–60 years: 3.2%, 61–65 years: 3.7%, 66–70 years: 6.9%, 71–75 years: 12.1% and >75 years: 14.1%). In logistic regression, ejaculatory dysfunction was correlated with age, history of prostate cancer, depressive syndrome, the individual LUTS of leaking during sex, and urgency with fear of leaking.

The ICS-BPH study enrolled a cohort of 1271 patients aged over 45 years from 12 countries with BPH-related LUTS. Four hundred and twenty-three patients were selected as a control group. All patients were asked to complete a questionnaire. No differences were found between the test group and the control group. Erectile and ejaculatory disorders were correlated with the LUTS severity. Ejaculatory dysfunction and painful ejaculations were found in 47% and 5% of cases in the test group [11].

Overall, BPH-related LUTS, are associated with either absence or reduction in ejaculatory volume. If this disorder is correlated with age and LUTS severity, the functional discomfort is not. Although ejaculatory pain or discomfort are less prevalent than other ejaculatory dysfunctions, their consequence on patients’ quality of life is significant.

### 3.4. Anatomic Rational for Ejaculatory Disorders in BPH

From an embryological point of view, prostate and ejaculatory ducts come from different structures; mesonephric duct for the ejaculatory ducts and primitive urogenital sinus for the prostate [15]. Ejaculatory ducts are devoid of muscular cells. They undergo the contraction of the seminal vesicles, the vas deferens, and the prostatic smooth muscle during the emission phase. Their role is more that of transport than of production or contraction. Their intraprostatic path is forward, downward, and medially, but when they reach the colliculus seminalis, they diverge and end up distinctly in the prostatic urethra. The anatomical study of the ejaculatory ducts is closely related to the exploration of the mechanisms of retrograde ejaculation in post-operative BPH surgery. The knowledge of their anatomic passage is an essential asset for the preservation of the ejaculation after TURP. However, only one study conducted by Malalasekera et al. looked specifically at the path and the anatomical relationships of the ejaculatory ducts, from 6 cadaveric subjects over 50 years old [16]. The authors firstly described the peri-montanal zone: 7.5 mm laterally and 10mm proximally from the veru montanum. Considering all prostate sizes, the ejaculatory ducts progress through this zone in 95% of cases.

### 3.5. Impact of Medical Treatments

#### 3.5.1. Phytotherapy

Herbal extracts are used alone or in combination for moderate LUTS. The systematic review conducted by Bauer et al. did not find any ejaculatory changes under phytotherapy (Table 1) [17]. MacDonald et al. did not report the sexual outcomes under phytotherapy [18]. Finally, Debruyne et al. compared alpha-blockers and phytotherapy with a higher grade of ejaculation disorders in the alpha-blockers group [19].

#### 3.5.2. Alpha-blockers

Alpha-blockers (AB) are alpha-adrenergic receptor antagonists. They can be selective (Silodosin and Tamsulosin) and target alpha-1a receptors responsible for the relaxation of the prostate muscle, or non-selective (Alfuzosin) and target additionally alpha1b and 1d receptors with the main undesirable effect of orthostatic hypotension. Alpha-blocker medication may be accompanied by ejaculatory modifications, such as reduced ejaculate volume or anejaculation (Table 2) [20,21,22,23,24,25].

#### 3.5.3. 5-Alpha Reductase Inhibitors (5ARI)

5ARI blocks an enzyme that converts testosterone to active dihydrotestosterone (DHT). One of the effects of DHT is to increase NO. It can explain ejaculatory and erectile dysfunction under 5ARI. As one of them, Finasteride is associated with a 4% rate of ejaculatory dysfunction (Table 3) [26,27,28,29]. An association of medical therapies is found to increase the adverse events rate, also regarding men’s sexuality [30].

### 3.6. Impact of Surgical Treatments

#### 3.6.1. Standard Endoscopic Procedures

Trans-urethral resection of the prostate (TURP, monopolar or bipolar): Ejaculatory dysfunction rate after monopolar TURP ranged between 50 and 70% in a recent literature review, despite Muntener et al. not finding any post-operative erectile or sexual modifications [31,32,33]. Only a few studies have directly compared monopolar and bipolar TURP. Chen et al. did not find any ejaculatory modification after a 2-year postoperative follow-up, however, the study methodology revealed a lack of power (Table 4) [34].

Trans-urethral incision of the prostate (TUIP): This procedure is reserved for young patients with persisting LUTS despite well-conducted medical treatment and low prostate volume (<30 mL). It consists of an incision between the right lateral lobe and the prostatic mid lobe without tissue resection. Several randomized studies have compared TUIP and TURP: Riehmann et al. evidenced a lower rate of retrograde ejaculation in the TUIP group compared to the TURP group (35% versus 68%, *p* = 0.02), confirmed secondarily in long term results (21% of post-operative retrograde ejaculation rate) (Table 4) [33,35].

#### 3.6.2. Greenlight Laser Photo Vaporization of the Prostate (PVP)

The main benefit of PVP is the hemostasis improvement in patients at high risk of bleeding. GOLIATH study found non-inferiority of PVP compared to TURP for functional results (Table 5) [36].

#### 3.6.3. Simple Prostatectomy (Open- and Robot-Assisted)

A single prospective study evaluated sexual function before and after open simple prostatectomy [37]. Using ICS-BPH (ICS-sex), the authors assessed the patients’ overall sexuality, demonstrating no change in satisfaction with intercourse but a significant increase in sexual desire and overall sexual satisfaction. Ejaculatory function was not assessed individually. With the development of mini-invasive techniques, robot-assisted SP (RASP) has gained popularity with the possibility of urethral-sparing (us) techniques for ejaculatory preservation [38]. Recently, us-RASP has been compared prospectively with standard RASP, based on MHSQ-EjD [39]. Beyond similar functional outcomes, antegrade ejaculation was maintained in 81% in us-RASP group, versus 8.8% in control group (RASP) at 12-month follow-up (*p* < 0.0001) (Table 6).

#### 3.6.4. Anatomic Endoscopic Enucleation of the Prostate (AEEP)

AEEP is an endoscopic simple prostatectomy, using laser or bipolar energy, with varying results and setbacks. Holmium laser enucleation of the prostate (HoLEP, Lumenis©, San Jose, CA, USA) is a technique with the longest follow-up up to 7 years depending on the study. A 75% retrograde ejaculation rate has been reported with this technique [40]. A feasibility study evaluated the benefit of preserving supramontanal tissue during HoLEP, finding a 15% decrease in the rate of retrograde ejaculation [41]. Greenlight laser enucleation of the prostate (GreenLEP, Boston Scientific Corporation©) has been developed secondarily to PVP, with the ability to treat any prostate volume using the Greenlight laser. With the En-Bloc technique, a prospective cohort study reported an antegrade ejaculation rate of 1.2% at 12 months follow-up [42]. Another study using lobe-by-lobe techniques reported a 36% rate of retrograde ejaculation that was equivalent to the percentage of patients having sexual activity [43]. Thulium laser enucleation of the prostate (ThuLEP, Dornier©) has been presented as an alternative to HoLEP for prostate volumes greater than 80 mL. Only one study reports data relative to the EjD (MSHQ) in a 177 patients ‘cohort, with a reduction in ejaculate volume and an antegrade ejaculation rate of 11.9% at 8 months of follow-up [44]. Finally, Thulium fiber laser enucleation of the prostate has been recently introduced (ThuFLEP, IPG©). If the peri-operative outcomes and erectile function are comparable to other laser sources and TURP, to our knowledge, no dedicated results regarding the ejaculatory function are available [45,46,47]. Bipolar enucleation of the prostate (BEEP) includes several procedures, such as plasmakinetic enucleation of the prostate (PkEP), transurethral resection enucleation of the prostate (TUERP), bipolar plasma enucleation of the prostate (BPEP), transurethral vapor-enucleation resection of the prostate (TVERP), transurethral vapor-enucleation of the prostate (TVEP), and finally, bipolar enucleation of the prostate (BipoLEP) [48]. As for ThuFLEP, to date, there is no valid data regarding EjD (Table 7).

### 3.7. Interventional Radiology: Prostate Artery Embolization (PAE)

PAE is a minimally invasive procedure, performed under local anesthesia and offered in cases of LUTS resistant to medical treatment or in case of disabling side effects. PAE is indicated in patients reluctant to surgical treatment, with numerous co-morbidities, or who are willing to keep antegrade ejaculations. A prospective study of 32 patients did not find a case of retrograde ejaculation with a 7 months follow-up [49]. A randomized study comparing PAE and TURP evidenced retrograde ejaculation in 10% of cases of PAE versus 100% in the TURP group [50]. Those findings are confirmed by two different studies (Table 8) [51,52]. A recent study compared TURP and PAE after a 2-year follow up and evidenced that TURP had a greater improvement regarding IPSS and maximum urinary flow rate, and a greater reduction in postvoid residual urine [53].

### 3.8. New Surgical Therapies

#### 3.8.1. Rezum©

Rezum or “convective water vapor energy ablation” (NxThera©) uses vapor water as the source for adenoma ablation [54]. A prospective randomized controlled study (against TURP) concluded on the feasibility of this technique (significant improvement in functional scores of LUTS and urinary flow) with 2.9% anejaculation and decrease in ejaculate volume at three months, 0% at 1 year, without significant difference with the control group. The functional impairment score related to an erectile disorder was significantly improved (31%, *p* = 0.001). These results persisted after 4 years of follow-up, with a retreatment rate of 4.4% (Table 9) [55,56].

#### 3.8.2. Prostate Urethral Lift or Urolift©

The aim is to spread the prostatic lateral cheeks to help restore a free urethral canal for urination while maintaining antegrade ejaculations. A prospective randomized study presented efficient functional outcomes associated with lasting improvements in ejaculatory function confirmed at 5 years of follow-up [57,58] (Table 10). These results were confirmed by two monocentric recent publications [59,60].

#### 3.8.3. Aquablation©

Aquablation (Aquabeam, Procept BioRobotics Corporation©) uses a high flow rate of physiological saline to perform the resection of the peri-urethral part of the prostate. Consequently, the peri-montanal tissue can be preserved.

Plante and al compared Aquablation to TURP, reporting a significantly lower rate of anejaculation in sexually active patients (2% versus 41%, *p* = 0.0001 at 6 months postoperatively) [61]. Recently, Aquablation was also associated with a lower rate of retrograde ejaculation compared to TURP (10% versus 36%, *p* = 0.0003), with no change in ejaculatory function on the MSHQ self-questionnaire (Table 11) [62].

## 4. Discussion

### 4.1. Definition and Evaluation

Before treating BPH-related EjD, the practitioner needs to characterize these disorders. We present, in this work, all available and valid self-questionnaires. The number and possible misunderstanding of the questions limit this evaluation method. For example, questionnaires cannot differentiate retrograde ejaculation from anejaculation. These two terms are often pooled in the literature, especially when the evaluation method is not detailed (or a simple self-questionnaire filling). It would probably be more appropriate to use the expression: “absence of antegrade ejaculation” or “ejaculatory dysfunction “in such studies.

### 4.2. Medical Therapies

Figure 1 presents a decision tree according to patients’ ejaculatory willing in case of benign prostatic hyperplasia. Medical treatments for BPH are widely studied and their side effects (including EjD) sometimes lead to stopping the treatment. Thus, their impact on sexuality has been studied, with incertitude whether the medication deteriorates an existing disorder or induces new disorders. Anejaculation seems to prevail over retrograde ejaculation in the BPH-treated EjD. If phytotherapy was not associated with negative ejaculatory effects, alpha-blockers showed variable rates of ejaculatory dysfunction. Non-selective AB (alfuzosin) is associated with a low dysejaculation rate (1%) but their use is limited by a risk of induced arterial hypotension [20]. Selective alpha-blockers (tamsulosin and silodosin) are responsible for higher rates of ejaculatory dysfunction (4.5–28%) [23,25]. According to the available literature, this is rather a reduction in the ejaculate volume or even anejaculation rather than retrograde ejaculation. With 5ARI, the EjD rate is estimated in the range of 4 to 18% [26,28]. If the mechanism is well known for AB, available data suggest 5ARI also reduces the ejaculate, in association with erectile dysfunction. This is consistent with the common pathophysiological hypotheses between erectile and ejaculatory dysfunction. As mentioned before, randomized studies only assess the presence or absence of ejaculation binarily, therefore, the reported rate of ejaculate volume reduction is probably underestimated. It is believed that the effects are major in the first 12 months of treatment. Moreover, the initial promoted study for 5ARI only included 67 patients, with an absence of safety re-evaluation since [28]. Regarding therapeutic combinations (AB-5ARI dual therapy), which are regularly prescribed, the MTOPS study showed a two-fold higher rate of EjD compared to monotherapy. Confirmed by the CombAT trial, EjD is often associated with erectile dysfunction, thus increasing the sexual impact of those medications [30]. We recommend that practitioners discuss these elements with the patient before starting dual therapy. New therapies, such as new alpha-blockers are being evaluated to limit these sexual side effects. If some are in the clinical phase and others are still in animal models, none are FDA approved.

### 4.3. Surgical Management

Various surgical options are available for drug-refractory BPH-related LUTS, and they are commonly linked to retrograde ejaculations. As historical options, TURP is complicated by retrograde ejaculation in 70% and TIUP in less than 20% [33,35]. Initially, it was believed that retrograde ejaculation was linked to the resection of muscle fibers in the bladder neck. Then, it was discovered that the region of the colliculus seminalis was essential in antegrade ejaculation by Gallizia in 1972. Then, operative techniques began to develop trying to preserve this region. Either way, preservation of sus-montanal tissue is not specific to one or the other surgical technique but can be applied to almost all procedures. To date, this is the only certainty for the preservation of antegrade ejaculations. When using montanal-sparing approaches, the antegrade ejaculation rate is 90.8%, 46.2%, and 86.6% for TURP, HoLEP, and PVP, respectively [65]. Despite its attractiveness, we currently lack benchmarks for its realization in clinical practice, especially since the resection volume does not seem to influence the incidence of postoperative retrograde ejaculations. Malalasekera was proposed to preserve the perimontanal tissue of 15 × 10 mm [16]. However, none of the anatomical subjects presented BPH, or anatomical median lobe, which consists of a major limitation. Therefore, we do not know yet whether the ejaculatory ducts present the same anatomical relationships in BPH, with or without a median lobe. This limitation is confirmed by Kim et al., who compared two AEEP techniques by HoLEP, one preserving 1 cm of tissue above the veru montanum, the other without preservation [41]. This study did not show a significant difference in the preservation of retrograde ejaculations (46% vs. 26% of preservation of normal or reduced ejaculation, *p* = 0.2). The low number of patients in this study may explain the lack of significance. To our knowledge, HOLEP is the only laser AEEP for which the preservation of the sus-montanal tissue has been studied.

Regarding GreenLEP, the literature data are discordant: from 99% to 36% of postoperative retrograde ejaculations, but 100% of patients were sexually active postoperatively [42,43]. This discrepancy is probably related to a lack of landmarks on the place of onset of enucleation above the veru montanum. Consequently, we can only conclude that sexuality is not altered after GreenLEP. ThuLEP and ThuFLEP, as recently introduced laser sources, present significantly lower short-term functional results than HoLEP on all parameters, however, long-term studies will state whether this laser source is adequate to soft tissue application [65]. The Thulium fiber laser is a priori the best to avoid unwanted diffusions, but only one work compared ThuFLEP to HoLEP, without any difference on functional outcomes. Several techniques have been developed to help surgeons preserve perimontal tissue such as augmented reality (3D MRI image fusion with the endoscope optics) or urethra-sparing RASP. Porpiglia et al. recently reported an 81% rate of maintained antegrade ejaculation when using the us-RASP (92 patients) technique, significantly higher than in the RASP control group (92 patients), in patients with median 140 cc prostate volumes [39]. In multivariable analysis, the ejaculation recovery was associated with younger age and the absence of urethral infection at 3 months, and only with younger age at 12 months. Therefore, RASP with us-techniques may compete with AEEP, even in prostate volumes lower than 100 cc.

### 4.4. New Surgical Therapies

Rezum©, Urolift©, and Aquablation© showed promising functional and sexual results in the available short- and mid-term studies. However, those pilot studies need to be confirmed in larger cohorts and long-term results. Before making a paradigm change, we could see those three options as an opportunity for younger patients with moderate LUTS and who are willing to keep antegrade ejaculations. Consequently, they could take place between medical therapies and surgical options but with a high retreatment rate: from 4.4% to 11% at 4 years for Rezum©, 13.6% at 7 years for Urolift©, 4.3% at 2 years for Aquablation© [56,60,62]. Patients will need adequate information before any surgical procedure, as the post-operative retreatment results have not been studied in dedicated studies.

Performed by interventional radiologists, PEA can be an option to treat moderate LUTS, with the absence of general anesthesia. Its functional results are inferior to TURP with a significant rate of retreatment (5% before 1 year, 15–20% after 1 year) while the ejaculatory results are variable (1–24.1% of retrograde ejaculations) [49,52]. Patients must be aware of the rare but major side effects (bladder necrosis, rectal hemorrhage) with PAE, even in the case of the PerFected embolization technique and because of the anatomical variations of the prostate arterial vascularization.

## 5. Conclusions

The prostate gland is a crossroad between the urinary and the seminal tract. Consequently, the specific clinical evaluation of the ejaculatory function is primordial, moreover, in patients with BPH-related LUTS, using a dedicated questionnaire. Understanding the causes of treated or untreated BPH-related ejaculatory disorders, based on physiological and anatomical consideration, must be developed in further ex vivo studies. Medical therapies have shown various ejaculatory modifications: reduction in ejaculate volume or anejaculation. Surgical procedures gather historical and recent endoscopic management, in which the preservation of the peri-montanal tissue seems essential for an ejaculation-sparing procedure. Us-RASP may overcome AEEP on this specific aspect. New surgical therapies are reshaping the place of surgical procedures in BPH, defining patients willing to preserve the ejaculatory function, rather than the LUTS intensity.

## Figures and Tables

**Figure 1 jcm-10-05788-f001:**
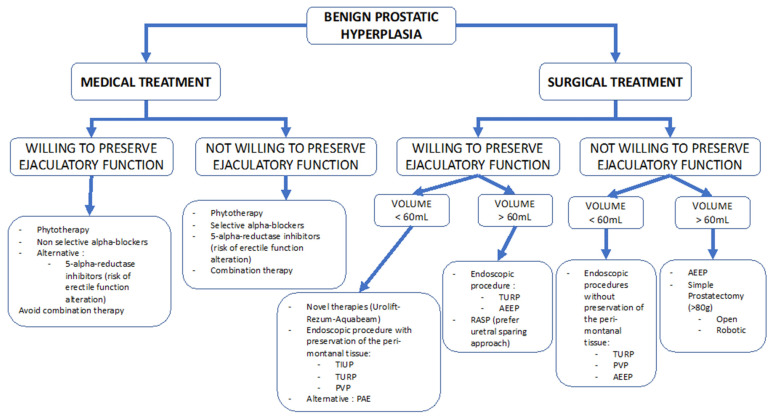
Decision tree according to the ejaculatory function patients’ demand and treatment in case of benign prostatic hyperplasia. *TUIP: transurethral incision of prostate; TURP: transurethral resection of prostate; PVP: photovaporization of prostate; PAE: prostate artery embolization; AEEP: anatomical endoscopic enucleation of the prostate; RASP: robot-assisted simple prostatectomy*.

**Table 1 jcm-10-05788-t001:** Impact of phytotherapy on ejaculatory status.

Reference	Aim	Study Design	Main Results	EjD Results
MacDonald R, et al., BJU Int. June 2012 [18]	To estimate the effectiveness and harms of Serenoa repens monotherapy in the treatment of lower urinary tract symptoms (LUTS) consistent with benign prostatic hyperplasia	Systematic review 17 randomised study, phytotherapy vs. placebo	Serenoa repens therapy does not improve LUTS or Q (max) compared with placebo	Not studied
Bauer HW et al., MMW Fortschr Med. 24 June 1999 [17]	To evaluate the efficacy of Saw palmetto fruit on urinary function	placebo-controlled double-blind study. Moderate-term study (6 months)	Statistically significant improvement of IPSS with Serenoa repens therapy (37% improvement) vs. placebo (14%)	No ejaculatory changes under phytotherapy
Debruyne et al., Eur Urol. May 2002 [19]	To assess the equivalent efficacy of Permixon and tamsulosin.	Prospective, double-blind randomized trial	no differences were observed in either irritative or obstructive symptom improvements after 1-year follow-up	ejaculation disorders occurred more frequently in the tamsulosin group (4.2% vs. 0.6% in Permixon group *p* = 0.001).

**Table 2 jcm-10-05788-t002:** Impact of unselective and selective alpha-blockers on the ejaculatory status.

Reference	Aim	Study Design	Main Results	EjD Results
Roehrborn CG et al., BJU Int. August 2003 [20]	To examine the efficacy and safety of a once-daily formulation of alfuzosin	Prospective randomized, double-blind, placebo-controlled 3-month study	Significant improve of IPPS score with alfuzosin vs. placebo −6.0 (5.1) vs. −4.2 (5.7) with placebo (*p* < 0.005) and the PFR, by + 2.3 (3.8) vs. + 1.1 (3.1) mL/s with placebo (*p* < 0.001)	Rare sexual adverse events with alfuzosin (impotence, 1.5%; ejaculation failure, 0.6%)
Van Moorselaar et al., BJU Int. March 2005 [21]	To assess the effect on sexual function of alfuzosin 10 mg once daily	Prospective, observationnal, real life practice study	alfuzosin significantly improved the total IPSS (−6.1, −32%)	Significant improvements in weighted scores related to reduced rigidity of erection (−0.5), reduced amount of ejaculate (−0.4) and pain/discomfort on ejaculation (−1.2, all *p* < 0.001) over baseline
Elhilali et al., BJU Int. March 2007 [22]	To assess the 2-year efficacy and safety of alfuzosin 10 mg once daily	Prospective, observationnal, real life practice study	total IPSS improved by 7 points (−38.5%) from baseline (*p* < 0.001)	Ejaculatory disorders were uncommon (0.3%)
Kobayashi et al., J Sex Med. September 2008 [23]	To evaluate the effect of silodosin on ejaculatory function of normal volunteers.	double-blind, placebo- controlled, randomized, crossover design N:15	100% anejaculation 0% retrograde ejaculation	100% anejaculation 0% retrograde ejaculation
Bozkurt et al., Urology. May 2015 [24]	To evaluate the sexual side effects including ejaculation after silodosin treatment in potent men with regular sexual activity	Prospective cohort N:30	Na	90% of impaired ejaculation
Chapple et al., Eur Urol. March 2011 [25]	To test silodosin’s superiority to placebo and noninferiority to tamsulosin	multicenter double-blind, placebo- and active-controlled parallel group study	IPSS total score with silodosin and tamsulosin was significantly superior to that with placebo (*p* < 0.001)	14% Anejaculation

**Table 3 jcm-10-05788-t003:** Impact of 5-alpha reductase inhibitors and associations with the ejaculatory status.

	Reference	Aim	Study Design	Main Results	EjD Results
5ARI	Fwu et al., J Urol. June 2014 [26]	To examine the effects of doxazosin, finasteride and combined therapy on sexual function	Multicenter, randomized, double-blind, placebo controlled	Slight worsening of ejaculatory function with finasteride and combined therapy compared with men on placebo. no significant difference in men assigned to doxazosin alone compared to placebo.	Non evaluated
	McVary et al., J Urol. May 2011 [27]	To revise the 2003 version of the American Urological Association’s (AUA) Guideline on the management of benign prostatic hyperplasia	Systematic review		Ejaculatory dysfunction of 4% (against 1% for the placebo) with finasteride
	McClellan et al., Drugs. 1999 [28]	Review of finasteride use in male pattern hair loss	Phase III		3.8% sexual function disorders (*p* < 0.041) −1.8% discreased libido −1.2% ejaculation disorder −1.3% erectile dyfunction
	Roehrborn et al., Urology. Sept 2002 [29]	To study the efficacy and safety of dutasteride	Randomized, double-blind, placebo controlled	Decrease in AUA-SI of 4.5 point at 24 months (*p* < 0.01)	2.2% ejaculation disorder (*p* < 0.01)
Associations	Roehrborn et al., J Urol. February 2008 [30]	To evaluate if combination therapy with dutasteride and tamsulosin is more effective than either monotherapy alone for improving symptoms and long-term outcomes in men with moderate to severe lower urinary tract symptoms and prostatic enlargement	Prospective, multicenter, randomized, double-blind, parallel group study	Significantly greater improvements in urinary symptoms with combinaison versus single therapy	Significant increase in drug related adverse events with combination therapy vs. monotherapies (×4)

**Table 4 jcm-10-05788-t004:** Impact of trans-ureteral resection of prostate and trans-ureteral incision of prostate on the ejaculatory status.

Reference	Aim	Study Design	Main Results	EjD Results
Riehmann et al., Urology. May 1995 [35]	To evaluate longer term effects of transurethral resection (TURP) and incision (TUIP) of the prostate in randomized patients.	Randomized, prospective study. Prostate < 20 cc	Decrease in obstructive symptoms in both groups (*p* < 0.034), no significant difference between the 2 groups.	68% retrograde ejaculation after TURP vs. 35% after TUIP (*p =* 0.02).
Marra et al., Int. J Urol. January 2013 [33]	To evaluate ejaculatory dysfunction in relation to benign prostatic hyperplasia surgery.	Systematic review; 42 randomized controlled trials comprising a total of 3857 patients were included.	66% retrograde ejaculation after TURP 21% after TUIP 41.9 after PVP 76.3% after HOLEP.	66% retrograde ejaculation after TURP 21% after TUIP 41.9 after PVP 76.3% after HOLEP.
Muntener et al., Eur Urol. August 2007 [32]	To evaluate the influence of TURP on erectile and ejaculatory function.	Prospective, multicenter, observational *N* 1014.	Significant decrease in ejaculatory function (*p* < 0.001) No significant difference of erectile function.	Significant decrease in ejaculatory function (*p* < 0.001) No significant difference in erectile function.
Chen et al., BJU Int. November 2010 [34]	To present 2-year follow-up data of a randomized clinical trial comparing bipolar transurethral resection in saline (TURIS) with monopolar transurethral resection of the prostate (TURP).	100 consecutive patients were randomized to TURIS or TURP.	Operative duration and resected tissue weight were similar between the groups significant improvements in IPSS and maximum urinary flow rates in both group.	50% retrograde ejaculation after TURP vs. 36% after TURIS (*p* = 0.52).

**Table 5 jcm-10-05788-t005:** Impact of photovaporization of prostate on ejaculatory status.

Reference	Aim	Study Design	Main Results	EjD Results
Bachmann et al., Eur Urol. May 2014 [36]	To evaluate the noninferiority of 180-W GL XPS (XPS) to TURP for International Prostate Symptom Score (IPSS) and maximum flow rate (Qmax) at 6 mo and the proportion of patients who were complication free.	Multicenter, Prospective randomised controlled trial. *N* 281.	Noninferiority of XPS to TURP for IPSS, Qmax, and complication-free proportion.	63% retrograde ejaculation after TURP vs. 65% after PVP.

**Table 6 jcm-10-05788-t006:** Impact of simple prostatectomy on ejaculatory status.

Reference	Aim	Study Design	Main Results	EjD Results
Gacci et al., BJU Int. February 2003 [37]	To evaluate urinary symptoms, sexual dysfunction and quality of life in patients with benign prostatic hypertrophy (BPH) before and after open prostatectomy	Monocentric Prospective *N* 60	Significant improvement in obstructive (mean 9.68–3.38) and irritative symptom (6.70–3.06), and quality-of-life scores (3.41–1.34)	No significant difference before and after SP concerning erectile and orgasm function
Porpiglia et al., Eur Urol. September 2020 [39]	To evaluate the efficacy of urethral-sparing robotic-assisted simple prostatectomy technique (usRASP) in obtaining effective deobstruction and maintaining anterograde ejaculation	Monocentric Prospective (retrospective control group) *N* 92.	Same perioperative and urinary functional outcomess in both groups	81% antegrade ejaculation in usRASP vs. 8.8% in RASP group

**Table 7 jcm-10-05788-t007:** Impact of anatomic endoscopic enucleation of Prostate on the ejaculatory status.

Reference	Aim	Study Design	Main Results	EjD Results
Welliver et al., Urol Clin North Am. August 2016 [31]	To consider potential pathophysiologic causes of dysfunction with treatment of LUTS due to BPH and attempts to critically review the available data to assess sexually related AEs.	Literature review		75% retrograde ejaculation
Wilson et al., Eur Urol. September 2006 [40]	To compare holmium laser enucleation of the prostate (HoLEP) with transurethral resection of the prostate (TURP) for treatment of men with bladder outflow obstruction (BOO) secondary to benign prostatic hyperplasia with a minimum of 24-month follow-up.	Randomized prospective trial *N* 61	HoLEP group: shorter catheter times and hospital stays; more prostate tissue retrieved; At six months, HoLEP was urodynamically superior to TURP in relieving BOO. No difference à 24 months (AUA, Qmax)	75% retrograde ejaculation in HOLEP; 62% in TURP
Kim et al., Int J Impot Res. February 2015 [41]	To explore the effectiveness of ejaculatory hood sparing technique to Holmium laser enucleation of the prostate (HoLEP) for ejaculation preservation	Prospective, controlled	Ejaculation preservation was 46.2% in the EH-HoLEP group and 26.9% in the conventional-HoLEP group (*p* = 0.249)	Ejaculation preservation was 46.2% in the EH-HoLEP group and 26.9% in the conventional-HoLEP group (*p* = 0.249)
Huet et al., Urology. Sept 2019 [42]	To evaluate the impact of Greenlight 180W photoselective vaporization of the prostate (PVP) and endoscopic enucleation of the prostate (GreenLEP) on ejaculatory and erectile functions.	Prospective, monocentric *N* 440	Antegrade ejaculation in 26.9% in the PVP group vs. 1.2% in the GreenLEP group at 12 months (*p* < 0.001)	Antegrade ejaculation in 26.9% in the PVP group vs. 1.2% in the GreenLEP group at 12 months (*p* < 0.001)
Bajic et al., Urology. Sept 2019 [43]	To present outcomes of a simplified GreenLight laser enucleation of the prostate (GreenLEP) technique and to inform urologists considering incorporation of enucleation into their practice.	Monocentric, prospective consecutive GreenLEPs by a single surgeon *N* 108	Significant improvements at 3 months in Qmax (237%, *p* < 0.01), in IPSS (−64%, *p* < 0.01), in postvoid residual (−83%, *p* < 0.01)	100% of retrograde ejaculation in patient with sexual activity (36%)
Saredi et al., Urol Int. 2016 [44]	To test the impact of Thulium laser enucleation of the prostate (ThuLEP) on erectile and ejaculatory functions, on lower urinary tract symptoms and on quality of life (QoL).	Monocentric, prospective *N* 177	Decrease in IPSS (*p* < 0.0001)	No difference in erectile function (IIEF) before and after surgery Reduction in ejaculation (*p* < 0.0001) 11.86% of antegrade ejaculation at 8 months
Enikeev et al., Int Urol Nephrol. November 2019 [47]	To perform a comparative analysis of en bloc and two-lobe techniques for holmium laser enucleation of the prostate (HoLEP) and thulium fiber laser enucleation of the prostate (ThuFLEP).	Retrospective *N* 1115	Mean surgery times (68.8 ± 30.6 min vs. 67.4 ± 30.1 min; *p* = 0.604) and enucleation rates (1.9 ± 0.74 g/min vs. 1.9 ± 0.69 g/min; *p* = 0.217) were comparable	No evaluation of ejaculatory function

**Table 8 jcm-10-05788-t008:** Impact of prostate artery embolization on the ejaculatory status.

Reference	Aim	Study Design	Main Results	EjD Results
Amouyal et al., Cardiovasc Intervent Radiol. Mar 2016 [49]	To report experience and clinical results on patients suffering from symptomatic BPH, who underwent PAE aiming at using the PErFecTED technique.	Single-center retrospective open label *N* 32	Mean IPSS decreased from 15.3 to 4.2 (*p* = 0.03), mean QoL from 5.4 to 2 (*p* = 0.03), mean Qmax increased from 9.2 to 19.2 (*p* = 0.25)	No retrograde ejaculation
Salem et al., Urology 2018 [51]	To evaluate the safety and efficacy of prostate artery embolization (PAE) for lower urinary tract symptoms (LUTS) attributed to benign prostatic hyperplasia	Prospective, single-center, open-label *N* 45	At 1 month, improvements in IPSS (23.6 ± 6.1 to 12.0 ± 5.9, *p* < 0.0001), QoL (4.8 ± 0.9 to 2.6 ± 1.6, *p* < 0.0001), Q_max_ (5.8 ± 1.0 to 12.4 ± 6.8, *p* < 0.0001). At 3 months, there were improvements in IPSS (10.2 ± 6.0, *p* < 0.0001), QoL (2.4 ± 1.6, *p* < 0.0001) and Q_max_ (15.3 ± 12.3, *p* < 0.0001). At 6 months, there were improvements in IPSS (11.0 ± 7.6, *p* < 0.0001) and QoL (2.3 ± 1.7, *p* < 0.0001). At 1 year, there were improvements in IPSS (12.4 ± 8.4, *p* < 0.0001) and QoL (2.6 ± 1.6, *p* < 0.0001).	No adverse effects on erectile function or sexual health
Ray et al., BJU Int. August 2018 [52]	To assess the efficacy and safety of prostate artery embolization (PAE) for lower urinary tract symptoms (LUTS) secondary to benign prostatic hyperplasia (BPH) and to conduct an indirect comparison of PAE with transurethral resection of the prostate (TURP)	Multicenter *N* 305	Median 10-point IPSS improvement from baseline at 12 months post-procedure	24.1% retrograde ejaculation rate for EAP against 47.5% for RTUP

**Table 9 jcm-10-05788-t009:** Impact of Rezum procedure on the ejaculatory status.

Reference	Aim	Study Design	Main Results	EjD Results
McVary et al., J Urol. May 2016 [54]	To evaluate the efficacy ok REZUM versus placebo	Multicenter, randomized, controlled study *N* 197	IPSS was reduced by 11.2 ± 7.6 in REZUM and 4.3 ± 6.9 in control (*p* < 0.0001)	No substantial decrements to erectil or ejaculatory function
McVary et al., J Sex Med. 2016 [55]	To determine whether water vapor thermal therapy would significantly improve lower urinary tract symptoms secondary to benign prostatic hyperplasia and urinary flow rate while preserving erectile and ejaculatory functions.	Multicenter, randomized, controlled study	IPSS and peak flow rate were significantly superior to controls at 3 months and throughout 1 year (*p* < 0.0001).a	0 de novo erectile dysfunction after REZUM IIEF was not differente between baseline and at 1 year. Ejaculatory bother score improved 31% over baseline (*p* = 0.0011).
McVary et al., Urology. 2019 [56]	To report 4-year outcomes of the randomized controlled trial of water vapor thermal therapy for treatment of moderate to severe lower urinary tract symptoms due to benign prostatic hyperplasia.		Lower urinary tract symptoms were significantly improved within ≤3 months after thermal therapy and remained consistently durable (International Prostate Symptom Score 47%, quality of life 43%, Qmax 50%, Benign Prostatic Hyperplasia Impact Index 52%) throughout 4 years (*p* < 0.0001)	No disturbances in sexual function were reported.

**Table 10 jcm-10-05788-t010:** Impact of Urolift procedure on the ejaculatory status.

Reference	Aim	Study Design	Main Results	EjD Results
Roehrborn et al., Can J Urol. Jun 2015 [57]	To report the three year results of use of the Prostatic Urethral Lift	Prospective, multi-center, randomized, blinded, sham control	IPSS improvement of 88% at three month 41.1% at 3 years	No de novo erectile or ejaculation dysfunction
Roehrborn et al., Can J Urol. Jun 2017 [58]	To report the five year results of use of the Prostatic Urethral Lift	Prospective, multi-center, randomized, blinded, sham control	IPSS improvement of 88% 41.1% at 3 years 36% at 5 years Surgical retreatment: 13.6% over 5 years	No de novo erectile or ejaculation dysfunction
Beurrier et al., Prog Urol. Jul 2015 [59]	To report the results of UroLift implants after a 2-year experience in the technique	Prospective monocentric *N* 23	Median IPSS and IPSS-QoL were improved significantly (11 [1–27] and 2 [0–6], *p* < 0.0001) No significant improved in Qmax	No patient reported retrograde ejaculation or worsened erectile function
Userovici et al., Prog Urol. Mar 2020 [60]	To report the results of Urolift^®^ system in our center after 7years experience.	*N* 40	At 3 months IPSS and IPSS-QdV were significantly improved (8 [4–11] vs. 20 [17–24]; *p* < 0.0001 and 2 [1,2] vs. 5 [4–6]; *p* < 0.0001).	MSHQ-EjD and IIEF5 were not modified (respectively 13 [11–14] vs. 12 [9–13]; *p* = 0.69 and 21 [18–23]; *p* = 0.13)

**Table 11 jcm-10-05788-t011:** Impact of Aquablation procedure on the ejaculatory status.

Reference	Aim	Study Design	Main Results	EjD Results
Plante et al., BJU Int. Apr 2019 [61]	To test the hypothesis that aquablation would have a more pronounced benefit in certain patient subgroups	Double-blind, multicentre prospective randomized controlled trial		Anejaculation 2% with aquablation vs. 41 with RTUP at 6 months (*p* < 0.0001)
Gilling et al., Adv Ther. Jun 2019 [62]	To compare 2-year safety and efficacy outcomes after Aquablation or transurethral resection of the prostate (TURP) for the treatment of lower urinary tract symptoms related to benign prostate hyperplasia	Prospective, randomised Blinded follow up *N* 181	IPSS simproved by 14.7 in Aquablation and 14.9 in TURP (*p* = 0.8304, 95% CI for difference—2.1–2.6 points)	Anejaculation 10% with aquablation vs. 36% with RTUP (*p* = 0.0003). No change in ejaculatory function on the MSHQ self-questionnaire with aquablation
Hwang et al., Cochrane Database Syst Rev. 2019 [63]	To assess the effects of Aquablation for the treatment of lower urinary tract symptoms in men with benign prostatic hyperplasia	Systematic review	Similar improvement in urologic symptom scores to TURP (mean difference (MD) −0.06, 95% confidence interval (CI) −2.51 to 2.39	No difference in IIEF before and after aquablation less ejaculatory dysfunction than TURP (MSHQED)
Bhojani et al., Urology. 2019 [64]	To report 12-month safety and effectiveness outcomes of the Aquablation procedure for the treatment of men with symptomatic benign prostatic hyperplasia (BPH) and large-volume prostates.	*N* 101	IPSS improved from 23.2 at baseline to 6.2 at 12 months (*p* < 0.0001)	Antegrade ejaculation was maintained in 81% of sexually active men

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
