# Peer review of "Ejaculations and Benign Prostatic Hyperplasia: An Impossible Compromise? A Comprehensive Review"

_jcm, 2021, doi:10.3390/jcm10245788_

Round 1

Reviewer 1 Report

The paper goes through litterature regarding Benign prostatic hypertrofia and ejaculatory modifications which an horonable effort because the area is not well elucidated. The paper is well written but ends up being quite mundane is simply reciting the litterature. I have some suggestions for emprovements. 

1) The authors are advised to include a recommendation for treatment in form of a decision three. It will be important that the readers gets a a few recommendations out of the review. 

2) The paper ends up taking a lot about the ejaculatory disordes and not so much about BHP. This should either be better balanced or the title changed to reflex the content

2) The authors are advised to present a figure caputing the main conclusions. The many tables gets a little to much of one thing for the readers. 

Author Response

Comment 1 :The paper goes through litterature regarding Benign prostatic hypertrofia and ejaculatory modifications which an horonable effort because the area is not well elucidated. The paper is well written but ends up being quite mundane is simply reciting the litterature. I have some suggestions for emprovements. 

Response : We gratefully thanks Reviewer #1 for his general comment on our manuscript. We tried to take those in consideration, in order to improve the quality of the manuscript.

Comment 2 :  The authors are advised to include a recommendation for treatment in form of a decision three. It will be important that the readers gets a a few recommendations out of the review. 

Response : we totally agree with the reviewer’s comment about the necessity of a clear figure that summarizes our findings. We added Figure 1 according to the reviewer’s recommendation. “Figure 1 presents a decision tree according to patients’ ejaculatory willings in case of benign prostatic hyperplasia. » was also added in the discussion part to invite the reader to refer itself to it.

Comment 3:  The paper ends up taking a lot about the ejaculatory disordes and not so much about BHP. This should either be better balanced or the title changed to reflex the content

Response : This comment is impactful, thanks to the reviewer. We modified our title according to the recommendation : “Ejaculations and benign prostatic hyperplasia : an impossible compromise? a comprehensive review”

Comment 4: The authors are advised to present a figure caputing the main conclusions. The many tables gets a little to much of one thing for the readers. 

Response : we aknowledge the lack of clear message at the end of the discussion. We improved it by adding Figure 1 and citing this figure early in the discussion for the readers. We tried to propose less tables but the high amount of cited articles impact on the clarity of the table. We hope that the reviewer the figure 1 will fit with his recommendations.

Reviewer 2 Report

Dear Authors,

manuscript is well conceived and might be a valuable contribution to the field. Nevertheless, I have only a point which raise my concern

In methods you state that 55 studies (line 80) were included, while in results you report 66 (line 91) included studies

Would you please correct or explain the difference?

Author Response

Comment 1: manuscript is well conceived and might be a valuable contribution to the field. Nevertheless, I have only a point which raise my concern.

Response 1:  We thanks Reviewer #2 for his general comment on our manuscript. We tried to take his comment in consideration and modified the manuscript accordingly

Comment 2: In methods you state that 55 studies (line 80) were included, while in results you report 66 (line 91) included studies. Would you please correct or explain the difference?

Response 2 : we totally agree with the reviewer’s comment and we modified the manuscript accordingly : 66 included studies as mentioned in the revised version of our manuscript.

Round 2

Reviewer 1 Report

The authors have done a good job in revising the paper. I have no futher comments